# Enhancing RABASAR for Multi-Temporal SAR Image Despeckling through Directional Filtering and Wavelet Transform

**DOI:** 10.3390/s23218916

**Published:** 2023-11-02

**Authors:** Lijing Bu, Jiayu Zhang, Zhengpeng Zhang, Yin Yang, Mingjun Deng

**Affiliations:** 1School of Automation and Electronic Information, Xiangtan University, Xiangtan 411105, China; lijingbu@126.com (L.B.); zhangzhengpeng2004@126.com (Z.Z.); xtudmj@xtu.edu.cn (M.D.); 2School of Mathematics and Computational Science, Xiangtan University, Xiangtan 411105, China; yangyinxtu@xtu.edu.cn; 3National Center for Applied Mathematics in Hunan Laboratory, Xiangtan 411105, China

**Keywords:** multi-temporal SAR images, image denoising, non-local means filtering, wavelet transform, ratio image

## Abstract

The presence of speckle noise severely hampers the interpretability of synthetic aperture radar (SAR) images. While research on despeckling single-temporal SAR images is well-established, there remains a significant gap in the study of despeckling multi-temporal SAR images. Addressing the limitations in the acquisition of the “superimage” and the generation of ratio images within the RABASAR despeckling framework, this paper proposes an enhanced framework. This enhanced framework proposes a direction-based segmentation approach for multi-temporal SAR non-local means filtering (DSMT-NLM) to obtain the “superimage”. The DSMT-NLM incorporates the concept of directional segmentation and extends the application of the non-local means (NLM) algorithm to multi-temporal images. Simultaneously, the enhanced framework employs a weighted averaging method based on wavelet transform (WAMWT) to generate superimposed images, thereby enhancing the generation process of ratio images. Experimental results demonstrate that compared to RABASAR, Frost, and NLM, the proposed method exhibits outstanding performance. It not only effectively removes speckle noise from multi-temporal SAR images and reduces the generation of false details, but also successfully achieves the fusion of multi-temporal information, aligning with experimental expectations.

## 1. Introduction

Multi-temporal SAR images utilize various auxiliary data sources for cross-validation and information supplementation [1], allowing for the observation of surface changes. Multi-temporal SAR provides a more comprehensive and detailed view of surface and dynamic changes [2]. SAR not only overcomes the limitations of time and weather conditions in ground observation [3], specifically demonstrating the capability of all-weather, all-time, and the continuous observation of moving targets [4,5], but also exhibits a certain ability to penetrate vegetation, soil, and occlusions [6,7]. Given these unique advantages of SAR, its applications are extremely diverse. For example, Zhang et al. proposed a near real-time method for monitoring the progression of forest fires [8]. Fu et al. mapped mangrove species and elucidated their scattering characteristics to monitor the extent and health of mangroves [9]. However, the presence of speckle noise significantly affects the quality and resolution of SAR images [10]. Therefore, removing speckle noise has always been a key issue for the further processing and applications of SAR images. Speckle removal can be broadly categorized into two types: single-temporal SAR denoising and multi-temporal SAR denoising.

Methods for single-temporal SAR speckle removal can be broadly categorized into three main types: spatial domain filtering, transform domain filtering, and deep learning filtering [11]. Spatial domain filtering includes Lee filtering [12], Frost filtering [13], and Kuan filtering [14], among others. The performance of spatial domain filtering is highly affected by the size of the filtering window, as smaller windows may not effectively suppress noise, while larger windows may lead to the loss of image texture details during denoising [15]. Transform domain filtering commonly utilizes Fourier transform [16], wavelet transform [17], and other techniques. Deep learning filtering includes MRDDANet [18] and AGSDNet [19], among others. SAR images inherently contain speckle noise during the imaging process, making it impossible to obtain completely clean images [20]. Supervised learning requires a training dataset with clean images, but there are no clean images in SAR. Moreover, deep learning models lack interpretability [21], which restricts their further applications in certain scenarios where model explanations are needed.

With the continuous development of SAR satellite technology, SAR satellites are capable of capturing multiple images of the same target with shorter time intervals. Given the increasing demand for multi-temporal SAR speckle removal, several common methods have emerged. Lê et al. proposed a novel method for the temporal adaptive despeckling of multi-temporal SAR images [22]. Chierchia et al. proposed a despeckling algorithm for multi-temporal SAR images, utilizing the principles of block-matching and collaborative filtering [23]. The RABASAR [24] proposed by Zhao et al. in 2019, is one of the most remarkable frameworks for multi-temporal SAR image despeckling in recent years [25]. The key idea of the RABASAR lies in the utilization of ratio images, as they are more amenable to despeckling due to their better spatial stationarity. However, the RABASAR still has some limitations in terms of obtaining the “superimage” and the ratio image.

Regarding the method of obtaining the “superimage”, the RABASAR adopts a weighted averaging technique during its first step to generate the so-called “superimage”. This approach may cause significant information loss in the input of multi-temporal SAR images, contradicting the goal of enriching image information through multi-temporal SAR images. To address this issue, we propose utilizing the DSMT-NLM to obtain the “superimage”. Antoni Buades et al. introduced the NLM [26], which incorporates both local and global information. Its concept still represents a groundbreaking advancement. However, the algorithm still has limitations, which can be analyzed from two aspects. Firstly, the NLM itself has room for improvement. Within the target window, there may be pixels with low relevance to the center pixel, resulting in situations where the weighted similarity calculation assigns too low of a weight to the center pixel values of the target window, despite their close resemblance to the center pixel values of the sliding window. This leads to significant discrepancies between the updated pixel values and the true values when updating through weighting. Secondly, the application of the NLM to multi-temporal SAR images poses challenges. Existing research methods often utilize single-temporal SAR images for NLM despeckling, which may result in insufficient information. The presence of speckle noise caused by coherence effects in SAR imaging blurs the boundary between the speckle noise and image details, making it challenging to distinguish speckle noise from details in SAR images. Multi-temporal SAR images are designed to address this information deficiency. To address the above issues, we propose the DSMT-NLM. The DSMT-NLM will utilize directional segmentation to identify the window with the highest correlation as the target window, effectively overcoming the distortion at the image edges. Additionally, we will traverse all time-series windows using a sliding window to maximize the utilization of information from all multi-temporal SAR images, thus improving the despeckling performance. Lastly, by combining data from multiple information sources, we will perform information fusion, thereby supplementing the content and dimensions lacking in a single information source, enhancing the completeness and accuracy of SAR image information, and obtaining the “superimage”.

Regarding the approach to obtain the ratio image, during the experimental process of the RABASAR we observed a phenomenon: by only selecting one image of interest for ratio calculation with the “superimage”, we found that this approach resulted in the final image containing only the geographical information of the selected interest image. This processing approach contradicts the goal of using the rich information from multi-temporal data to generate the “superimage”. To address this issue, we improved the method of generating the ratio image. We adopted a WAMWT to fuse information from the multi-temporal SAR images, resulting in the creation of a superimposed image. Subsequently, we performed the ratio operation between the superimposed image and the “superimage” to generate the ratio image. This processing approach preserves the characteristics of the multi-temporal information, thus obtaining more accurate despeckling results.

The main contributions of this study are as follows:

We propose a DSMT-NLM to acquire a high-quality “superimage”.We employ a WAMWT to fuse information from multi-temporal SAR images, producing a superimposed image. Subsequently, we perform a ratio operation between the superimposed image and the “superimage” to generate the ratio image.We introduce a directional segmentation method to calculate the window with the highest correlation as the target window.By employing a sliding window to traverse all time-series images, we maximize the utilization of information from all temporal SAR images, significantly enhancing the despeckling effect.

## 2. Research Methodology

The flowchart of the whole framework is illustrated in Figure 1. The algorithm comprises the following steps:

Step 1: From the input multi-temporal SAR images, sequentially select each SAR image of the time series as the reference image.

Step 2: For the pixels to be despeckled in the reference image, perform directional segmentation within their neighborhood, resulting in eight directional windows: up, down, left, right, left-up, left-down, right-up, and right-down. Calculate the weighted average of pixels within each directional window to obtain their mean values. Then, utilize the correlation distance to calculate the relevance between the pixel mean values of each directional window and the pixels to be despeckled in the reference image. Among the eight directional windows, identify the one with the maximum relevance to the pixels to be despeckled in the reference image, and select that directional window as the target window.

Step 3: For each SAR image in the time series, including the selected reference image, set a search window. Calculate the similarity between the target window and the sliding windows within the search window to determine the weights of the center pixels in the sliding windows. Multiply each center pixel value of the sliding window by its corresponding weight, and then calculate the weighted average of these products. The resulting value is used to update the pixel to be despeckled in the target window. Repeat the above process for each pixel in the reference image, thus completing the despeckling process for the reference image. Similarly, repeat the above steps for other SAR images selected as reference images to achieve the despeckling for all reference images.

Step 4: Apply wavelet transform to each filtered reference image to decompose it into different frequency components. Merge these components and reconstruct the “superimage” through wavelet inverse transform. Perform MuLoG-BM3D filtering on the “superimage”. At the same time, follow the same fusion process for the unfiltered input images to obtain the superimposed image. Then, calculate the ratio image by performing the ratio operation between the superimposed image and the filtered “superimage”.

Step 5: Apply RuLoG filtering to the ratio image, and then perform the inverse transform to obtain the final image.

### 2.1. Preliminary Despeckling of Multi-Temporal SAR Images Based on Directional Segmentation

The original RABASAR utilizes a weighted average method to generate the “superimage”, which achieves initial despeckling and integrates the information from multi-temporal SAR images. However, this simple approach leads to unsatisfactory despeckling results and fails to fully exploit the abundant information features in multi-temporal SAR images. Moreover, the use of weighted averaging causes the smoothing of image texture, resulting in the severe loss of feature information. This contradicts the fundamental idea of utilizing multi-temporal SAR images to compensate for the insufficient information in single-temporal SAR images. Therefore, to address these issues, we propose utilizing the DSMT-NLM method to generate the “superimage”, as illustrated in Figure 2. The specific steps are detailed in this section.

#### 2.1.1. Selection of Target Window Based on Directional Segmentation

In the original NLM and its subsequent improvements, preserving edge details is often overlooked. This is because there are pixels in the center pixel block that have a low correlation with the center pixel, especially in the regions near the edges. This significantly interferes with the weight calculation between the target window and the sliding window, causing blurriness at the image edges and leading to the loss of edge details. To address this issue, our algorithm improves upon the NLM by introducing directional segmentation to guide the selection of the target window, thereby mitigating edge blurriness.

For each pixel (i,j) to be despeckled in the reference image, along with its corresponding neighborhood region $w_{ij}$, our algorithm adopts directional segmentation to obtain eight directional windows, denoted as $w_{ij}^{k}$, where k represents the eight directions: left-up, left-down, right-up, right-down, up, down, left, and right. The schematic diagram of the directional segmentation is illustrated in Figure 3.

By taking the weighted average of the pixel values within the directional window $w_{ij}^{k}$, we obtain the pixel value mean of the directional window. To provide a clearer explanation, let us take the left-up directional window $w_{ij}^{Left-up}$ as an example. Figure 4 shows the pixel points within $w_{ij}^{Left-up}$, denoted as $\left(a_{1},b_{1}\right)$, $\left(a_{2},b_{2}\right)$, $\left(a_{3},b_{3}\right)$, and (i,j). We assign the weights $w_{a_{1}b_{1}}$,$\;w_{a_{2}b_{2}}$,${\;w}_{a_{3}b_{3}}$, and $w_{ij}$ to the pixel values $p_{a_{1}b_{1}}$, $p_{a_{2}b_{2}}$, $p_{a_{3}b_{3}}$, and $p_{ij}$, respectively. After the weighted average calculation, we obtain the pixel value mean $M_{Left-up}$, as shown in Equation (1), where $N_{Left-up}$ represents the sum of the weights of each pixel point, as shown in Equation (2).
(1)$$M_{Left-up}=\frac{1}{N_{Left-up}}(p_{a_{1}b_{1}}\cdot w_{a_{1}b_{1}}+p_{a_{2}b_{2}}\cdot w_{a_{2}b_{2}}+p_{a_{3}b_{3}}\cdot w_{a_{3}b_{3}}+p_{ij}\cdot w_{ij})$$
(2)$$N_{Left-up}=w_{a_{1}b_{1}}+w_{a_{2}b_{2}}+w_{a_{3}b_{3}}+w_{ij}$$

Next, we calculate the correlation distance $R_{ij}^{k}$ between the pixel mean $M_{k}$ of each directional window and the pixel value to be despeckled $p_{ij}$ using Equation (3). Then, we select the directional window corresponding to the highest correlation distance as the target window $W_{1}$ for the pixel to be despeckled, as shown in Equation (4).
(3)$$R_{ij}^{k}=|p_{ij}-M_{k}|$$
(4)$$W_{1}={argmin}_{k}R_{ij}^{k}$$

Here, k∈{left-up, left-down, right-up, right-down, up, down, left, and right}.

#### 2.1.2. Despeckling of Multi-Temporal SAR Images Based on DSMT-NLM

In previous research on SAR image despeckling, the NLM was commonly used, but it was typically applied only to single-temporal SAR images. Due to the speckle noise in SAR images, the uniqueness of this speckle noise makes it difficult to accurately distinguish between the speckle noise and image details, leading to a blurring of the boundary between the speckle noise and image details, which increases the difficulty of speckle noise removal while preserving image details. The NLM is primarily designed for denoising single-temporal images and may have limited effectiveness in handling speckle noise. Compared to single-temporal SAR images, multi-temporal SAR images contain richer information about the scene, and the correlation information from multiple sequences can better estimate noise and preserve image details accurately. Therefore, in order to better distinguish between speckle noise and image details and provide more accurate noise estimation, we extended the principles of the NLM to better adapt it to multi-temporal SAR images. This specific method involves expanding the traditional NLM algorithm’s sliding window selection strategy from a single image to multiple images, allowing the sliding window to traverse all temporal SAR images. This extension allows the NLM to better utilize temporal information and achieve more accurate and reliable SAR despeckling.

In the search window of the multi-temporal SAR image, we select the sliding window $W_{2}$ and traverse it across all temporal SAR images. The size of the sliding window $W_{2}$ remains consistent with the target window $W_{1}$ selected in Section 2.1.1. The similarity between the target window $W_{1}$ and the sliding window $W_{2}$ is calculated using Equation (5). The similarity $S(W_{1},W_{2})$ is used to calculate the weight $w(W_{1},W_{2})$ corresponding to the center pixel value $p_{xy}$ of the sliding window $W_{2}$, as shown in Equation (6), where h is the smoothing parameter and T is the normalization coefficient. By multiplying $p_{xy}$ by the corresponding weight $w(W_{1},W_{2})$ and summing the values, then taking the average, we can update the pixel value $p_{ij}$ of the target window $W_{1}$ to the despeckled pixel value $\tilde{p_{ij}}$, as shown in Equation (8). We repeat this process for each pixel point of the reference image to obtain the despeckled reference image $C_{i}$.
(5)$$S(W_{1},W_{2})=\frac{1}{HW}\sum_{u=0}^{H-1}\sum_{v=0}^{W-1}{\left(W_{1}\left(u,v\right)-W_{2}(u,v)\right)}^{2}$$
where $W_{1}\left(u,v\right)$ represents the pixel value at point $\left(u,v\right)$ in the target window, and $W_{2}(u,v)$ represents the corresponding pixel value at point $\left(u,v\right)$ in the sliding window. Given that the sizes of the target window and sliding window are equal, the terms H and W mentioned here respectively denote the number of rows and columns in either the target window or the sliding window.
(6)$$w(W_{1},W_{2})=\frac{1}{T}exp(-\frac{S(W_{1},W_{2})}{h^{2}})$$

Here,
(7)$$T=\sum exp(-\frac{S(W_{1},W_{2})}{h^{2}})$$
(8)$$\tilde{p_{ij}}=\sum_{x,y}w(W_{1},W_{2})\cdot p_{xy}$$

#### 2.1.3. Weighted Average Information Fusion Method Based on Wavelet Transform

Due to the richer information content in multi-temporal SAR images, information fusion, which combines features from different sources to obtain more comprehensive and reliable information, is often necessary to obtain a “superimage.” Conventional approaches in previous research typically involve weighted averaging of multiple SAR images, which is a relatively simple form of information fusion. However, this method directly uses fixed weights to average the pixel values in the images, leading to the loss of rich texture details in the multi-temporal SAR images and a decrease in image clarity. To address these issues, we propose using a WAMWT to replace the conventional simple weighted averaging approach used in previous studies. This method utilizes wavelet transform for multi-scale decomposition, fully considering the frequency characteristics of SAR images and more finely processing information in different frequency ranges. By decomposing, fusing, and reconstructing multiple images, a more comprehensive “superimage” is synthesized, achieving more accurate information fusion. The specific steps of this method are as follows: First, the despeckled reference images $C_{i}$ are subjected to four-scale decomposition using the Daubechies 4 wavelet, obtaining different wavelet coefficients representing information in different frequency ranges, including the low-frequency sub-band (cA), horizontal high-frequency sub-band (cH), vertical high-frequency sub-band (cV), and diagonal high-frequency sub-band (cD), as shown in Equation (9).
(9)$$\left[{cA}_{i},{cH}_{i},{cV}_{i},{cD}_{i}]=dwt2[C_{i}\right]$$
where $C_{i}$ represents the input i-th reference image, and dwt2 is the function for the two-dimensional discrete wavelet transform.

For each wavelet sub-band, a weighted average is performed based on the corresponding weights, as shown in Equation (10).
(10)$${cA}_{fused}=\alpha_{1}\times {cA}_{1}+\cdot \cdot \cdot +\alpha_{i}\times {cA}_{i}\;{cH}_{fused}=\alpha_{1}\times {cH}_{1}+\cdot \cdot \cdot +\alpha_{i}\times {cH}_{i}\;{cV}_{fused}=\alpha_{1}\times {cV}_{1}+\cdot \cdot \cdot +\alpha_{i}\times {cV}_{i}\;{cD}_{fused}=\alpha_{1}\times {cD}_{1}+\cdot \cdot \cdot +\alpha_{i}\times {cD}_{i}$$

The weighted average of the wavelet sub-bands is then reconstructed through the inverse wavelet transform, specifically using the Daubechies 4 wavelet, resulting in the final “superimage” A, as shown in Equation (11).
(11)$$A=idwt2({cA}_{fused},{cH}_{fused},{cV}_{fused},{cD}_{fused})$$
where idwt2 represents the two-dimensional discrete inverse wavelet transform.

### 2.2. Residual Speckle Noise Removal Based on Ratio Image

After applying the DSMT-NLM, we obtained the “superimage”. This image has undergone initial despeckling and effective information fusion, but residual speckle noise still persists. To address this issue, the RABASAR employs the concept of a ratio image as its core approach. In the first section, we discussed the RABASAR’s method of selecting one SAR image of interest and generating a ratio image with the “superimage”. However, this approach fails to fully utilize the abundant information contained in the multi-temporal SAR images and contradicts the objective of extensively exploiting the multi-temporal information to generate the “superimage”. Therefore, we propose a new method, the weighted average based on wavelet transform, to replace the RABASAR’s approach of selecting only one SAR image of interest and generating a ratio image with the “superimage”. With this method, we can better utilize the information from the multi-temporal SAR images while maintaining the consistency of the previous step.

We applied the WAMWT to process the input multi-temporal SAR images, following a procedure similar to that described in Section 2.1.3, to obtain the superimposed image S. According to the RABASAR, by applying the MuLoG-BM3D filter to the “superimage”, we obtained the processed “superimage” [24]. Then, by performing a ratio operation between the superimposed image S and the processed “superimage” A, we obtained the ratio image τ, as shown in Equation (12).
(12)$$T=\frac{S}{A}$$

Next, we denoised the ratio image using the RuLoG algorithm [24]. Finally, we performed a restoration operation on the filtered ratio image by multiplying it with the despeckled “superimage” to obtain the final image, denoted as ${\hat{u}}_{t}$, as shown in Equation (13).
(13)$${\hat{u}}_{t}=A\cdot \hat{\tau }$$

## 3. Experiment

### 3.1. Evaluation Metrics

To objectively assess the effectiveness of the proposed framework, we employed several evaluation metrics to evaluate the experimental results, including the Structural Similarity (SSIM), Natural Image Quality Evaluator (Niqe), Correlation Coefficient (corrcoef), Signal-to-Noise Ratio (SNR), Peak Signal-to-Noise Ratio (PSNR), and the Equivalent Number of Looks (ENL). These evaluation metrics provide quantitative measures of image quality, noise level, and detail preservation ability, allowing for a comprehensive assessment of the algorithm’s performance.

#### 3.1.1. SSIM

SSIM takes into consideration the similarity in brightness, contrast, and structure to evaluate the resemblance between two images, as illustrated in the Equation (14).
(14)$$SSIM=\frac{\left(2\mu_{x}\mu_{y}+C_{1})(2\sigma_{xy}+C_{2}\right)}{\left(\mu_{x}^{2}+\mu_{y}^{2}+C_{1})(\sigma_{x}^{2}+\sigma_{y}^{2}+C_{2}\right)}$$

Here, x and y represent the two images. $\mu_{x}$ and $\mu_{y}$ are the pixel means of x and y, respectively. $\sigma_{x}$ and $\sigma_{y}$ are the pixel variances of x and y, respectively. $\sigma_{xy}$ is the pixel covariance between x and y. $C_{1}$ and $C_{2}$ are constants used to stabilize division to avoid division by zero in the denominator.

#### 3.1.2. Niqe

Niqe utilizes the statistical characteristics of images, such as contrast, brightness, and sharpness, to assess image quality. A lower Niqe score indicates higher image quality, making it a no-reference image quality assessment metric. The mathematical expression of Niqe is quite complex and will be omitted here.

#### 3.1.3. Corrcoef

Corrcoef is a function used to calculate the correlation coefficient between two sets of data. Mathematically, the correlation coefficient measures the degree of linear association between two sets of data. It ranges from −1 to 1. A correlation coefficient of 1 indicates a perfect positive correlation between the two sets of data.

#### 3.1.4. SNR

SNR is a metric used to evaluate the denoising effectiveness of an image. A higher SNR value indicates better image quality. It is calculated using Equation (15), where m and n represent the number of pixels along the length and width of the image, respectively. x(i,j) and y(i,j) represent the pixel values at location (i,j) in the original and filtered images, respectively.
(15)$$SNR=10{log}_{10}\frac{\sum_{i=1}^{m}\sum_{j=1}^{n}{x(i,j)}^{2}}{\sum_{i=1}^{m}\sum_{j=1}^{n}{\left[x\left(i,j\right)-y(i,j)\right]}^{2}}$$

#### 3.1.5. PSNR

PSNR is one of the most widely used evaluation metrics in image visual processing, which is utilized to represent the degree of image quality loss. The larger the PSNR value, the higher the image quality, indicating a smaller degree of distortion between two images and a higher degree of similarity, as shown in Equation (16).
(16)$$\mathrm{PSNR}=10\times \log_{10}\left[\frac{{\left(2^{n}-1\right)}^{2}}{MSE}\right]$$

Here,
(17)$$MSE=\frac{1}{n}\sum_{i=1}^{n}{\left(\hat{y_{i}}-y_{i}\right)}^{2}$$

$\hat{y_{i}}$ represents the final image after undergoing filtering, while $y_{i}$ refers to the input image without filtering.

#### 3.1.6. ENL

ENL is one of the crucial indicators for assessing image quality, representing a dimensionless measure. A larger ENL value indicates a smoother image with lower noise levels, implying more effective noise removal and less coherent speckle noise. ENL is defined as shown in Equation (18), where X denotes the set of sample points in a homogeneous region, $E_{X}$ represents the mean value of the homogeneous region, and $D_{X}$ represents the variance of the homogeneous region.
(18)$$ENL\left(X\right)=\frac{{\left(E_{X}\right)}^{2}}{D_{X}}$$

### 3.2. Experimental Analysis

To validate the effectiveness of the algorithm, experiments were conducted on simulated multi-temporal SAR images and real multi-temporal SAR images, separately.

#### 3.2.1. Analysis of Experiments on Simulated Multi-Temporal SAR Images

A certain aerial image was used as the clean image for simulating the multi-temporal SAR images, with an image size of 256 pixels × 256 pixels. To simulate the randomness of multi-temporal SAR image speckle noise, we applied speckle noise of different variances to the clean image. Specifically, three different noise variance values were selected: 0.04, 0.05, and 0.1. The applied speckle noise follows a uniform distribution with a mean of 0. By applying these speckle noises, we generated the simulated multi-temporal SAR images shown in Figure 5.

In order to assess the performance of our algorithm more accurately, we will employ four methods mentioned in the RABASAR, namely the arithmetic mean (AM), the denoised arithmetic mean (DAM), the binary-weighted arithmetic mean (BWAM), and the denoised binary-weighted arithmetic mean (DBWAM) [24], as comparative methods in our experiments, along with the Frost and NLM.

We compare the final image with the clean image and reference algorithms, as shown in Figure 6. It is worth noting that the conventional NLM and Frost are designed for despeckling individual images. To ensure comparability in our experiments, we combined the images from Sequence 1, Sequence 2, and Sequence 3 to create an image averaging. This image averaging was then used as the input for both the NLM and Frost algorithms.

From a visual perspective based on Figure 6, we observed that the performance of AM and BWAM in removing speckle noise was not satisfactory. Conversely, DAM, DBWAM, Frost, NLM, and the proposed method each demonstrated effective suppression. However, Frost, while removing speckle noise, caused the image to become blurred, resulting in no improvement in the clarity of features after speckle noise removal. Additionally, NLM exhibited over-smoothing, leading to a significant loss of terrain information. Moreover, the white features still retained visible granular speckle noise.

From the comparison of detail amplification in Figure 7, we observed that DAM and DBWAM generated pseudo-details, referring to striped textures that were not present in the clean image. Conversely, Frost, NLM, and the proposed method did not exhibit this phenomenon.

After analyzing the evaluation metrics in Table 1, it is evident that the BWAM, DBWAM, and proposed method have achieved relatively high scores in measuring the structural similarity between the final image and the clean image. This indicates their commendable performance in preserving the image structures. In terms of assessing image clarity and naturalness, it is noteworthy that the Niqe metric for the proposed method attains the lowest value. This implies its exceptional proficiency in maintaining image clarity and naturalness, signifying a superior image quality and enhanced detail clarity. Furthermore, when evaluating the strength and direction of the linear relationship between the final images and the clean images, the proposed method, along with the BWAM and DBWAM, demonstrates commendable metrics. This underscores their excellence in preserving linear relationships among image features.

Taking into account the comprehensive analysis of visual perception, detail amplification, and evaluation metrics, the proposed method excels in all three aspects, showcasing its remarkable capability for enhancing image quality. In contrast, other comparative methods each exhibit their own limitations.

#### 3.2.2. Analysis of Experiments on Real Multi-Temporal SAR Images

We selected two sets of real, spaceborne, multi-temporal SAR images as input data, each containing three temporal sequences. The detailed parameters of each multi-temporal SAR image set are presented in Table 2 and Table 3. Additionally, we performed registration processing on both sets of images to ensure their spatial alignment, as illustrated in Figure 8 and Figure 9.

From the experimental results in Figure 10 and Figure 11, we can observe that both the proposed algorithm and the comparative algorithms achieved certain results in removing speckle noise. However, for the AM and BWAM the removal of speckle noise was not thorough enough, resulting in noticeable speckle grains in the images. The Frost results in image blurriness and lower quality. The NLM suffers from noticeable over-sharpening of the image. In contrast, the proposed algorithm, DAM, and DBWAM achieved more effective suppression of speckle noise, leading to a significant reduction in speckle noise. Although the DBWAM and DAM achieved a certain level of speckle noise suppression, there were still faint traces of speckle noise in the detail regions, affecting the overall image quality. Conversely, the proposed algorithm effectively suppressed speckle noise both globally and locally, resulting in no noticeable speckle noise in the images and the preserving of edge details.

Through the horizontal comparison of the geographic information in Figure 12 and Figure 13, significant differences in geographic information among them are observed. Although the RABASAR was designed for multi-temporal SAR image filtering, it did not effectively perform information fusion, resulting in limited feature fusion and a diminished utilization of information value in multi-temporal images, consequently affecting subsequent data analysis. In contrast, the proposed algorithm achieved feature fusion across the sequences of images, maximizing the utilization of information from all image sequences, leading to a significant improvement in feature detail information.

Based on the analysis of the objective evaluation metrics (Table 4), the proposed algorithm achieved impressive results. The proposed algorithm produced images with higher quality, better visual fidelity, and minimal loss, effectively preserving image details. This is attributed to the directional segmentation of pixels to be denoised around the reference image, selecting the directional window with the highest correlation with the target window, which mitigates the influence of low-correlation pixels on weight calculation between the target and sliding windows, thereby enhancing the capability of coherent speckle noise removal while effectively preserving image details. Additionally, the proposed algorithm considered the characteristics of multi-temporal SAR images and achieved comprehensive information fusion. Unlike traditional non-local mean algorithms, the proposed algorithm traverses multi-temporal SAR images with a sliding window, efficiently utilizing spatial domain methods for multi-temporal information fusion, thereby maximizing the exploitation of information features in multi-temporal images. Therefore, compared to other methods, the proposed algorithm exhibits superior performance in multi-temporal SAR image processing.

## 4. Conclusions

In this paper, we proposed an algorithm, titled “Enhancing RABASAR for Multi-Temporal SAR Image Denoising through Directional Filtering and Wavelet Transform,” to address the challenge of speckle noise removal in multi-temporal SAR images. The proposed algorithm introduced a novel approach to obtain the “superimage”, referred to as DSMT-NLM. Additionally, we utilized a WAMWT to generate the superimposed image, which was then ratioed with the “superimage” to obtain the ratio image. Through subjective visual evaluation and objective performance metrics, we not only demonstrated the feasibility of the proposed approach but also showcased its superiority over the other methods. However, during the experiments on real multi-temporal SAR image II, we noticed that the image contrast of the experimental results did not reach the level of other comparative experiments, indicating a new challenge that we need to address. Despite the excellent performance of our algorithm in other aspects, the issue of insufficient contrast still requires further in-depth research and resolution. We acknowledge that this problem might stem from certain aspects or parameter settings of the algorithm. Therefore, in future research, we will focus on exploring and optimizing these aspects to achieve better contrast performance.

## Figures and Tables

**Figure 1 sensors-23-08916-f001:**
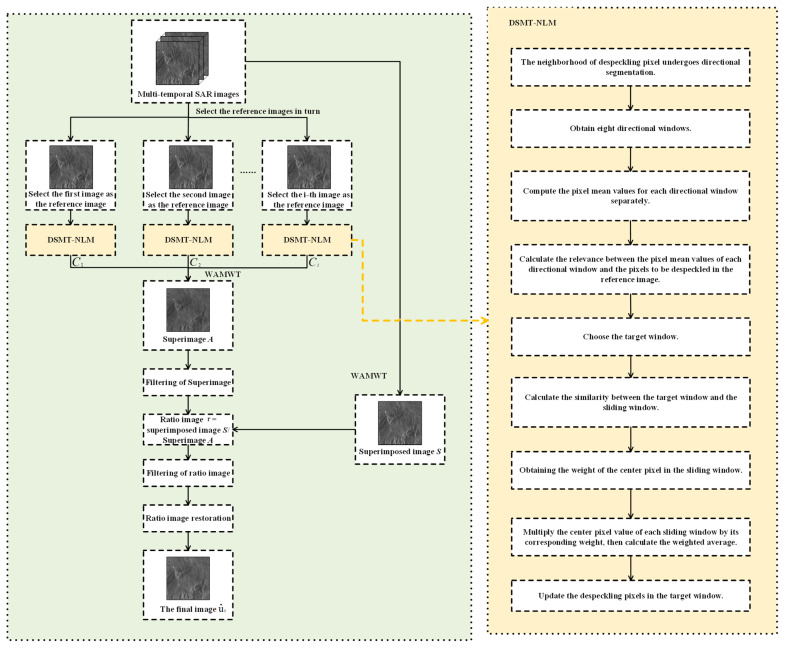
Flowchart of the whole framework.

**Figure 2 sensors-23-08916-f002:**
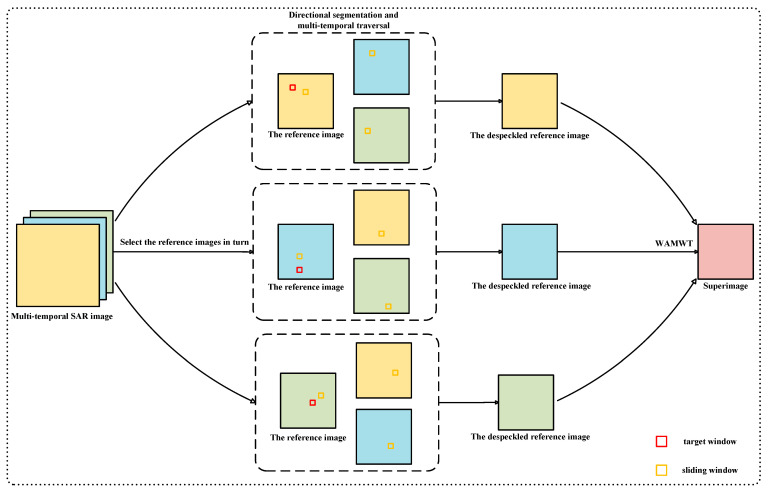
Illustrative diagram of the DSMT-NLM.

**Figure 3 sensors-23-08916-f003:**
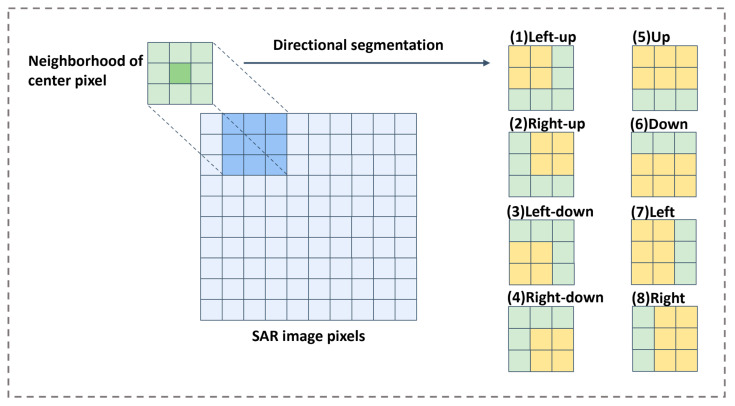
The schematic diagram of the directional segmentation.

**Figure 4 sensors-23-08916-f004:**
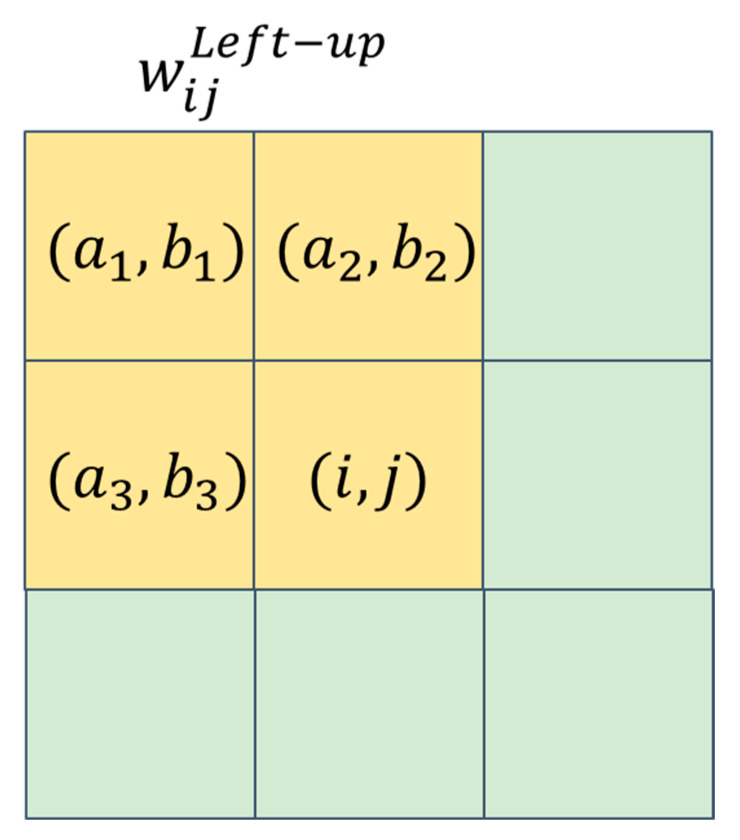
The schematic diagram of the directional window (left-up).

**Figure 5 sensors-23-08916-f005:**
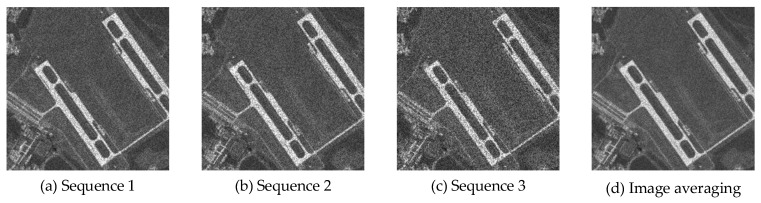
Untreated simulated multi-temporal SAR images.

**Figure 6 sensors-23-08916-f006:**
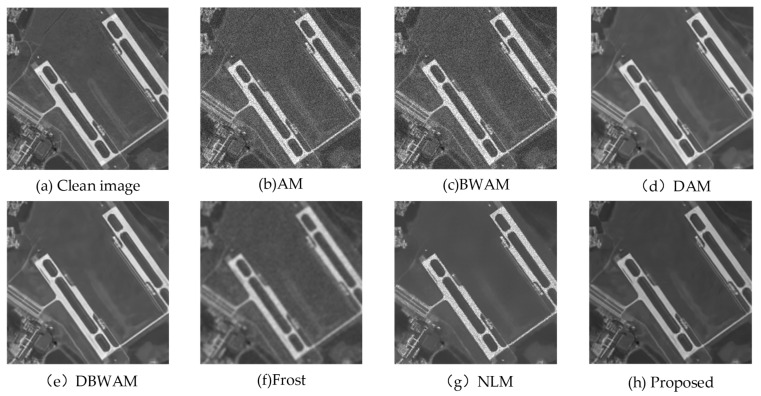
Experimental results.

**Figure 7 sensors-23-08916-f007:**
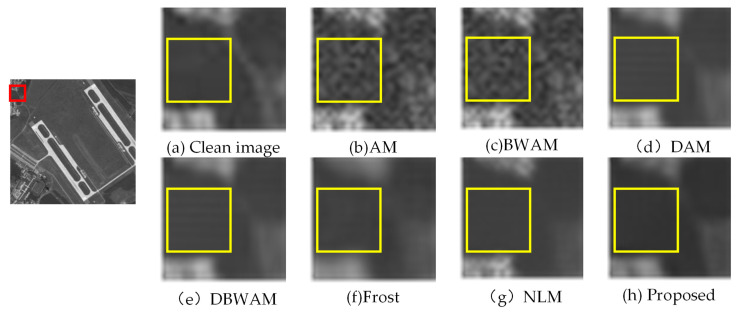
Comparison of detail amplification.

**Figure 8 sensors-23-08916-f008:**
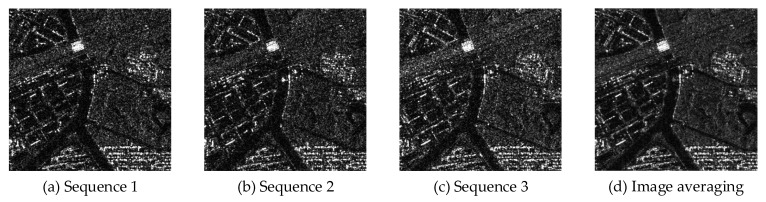
Untreated real multi-temporal SAR image (I).

**Figure 9 sensors-23-08916-f009:**
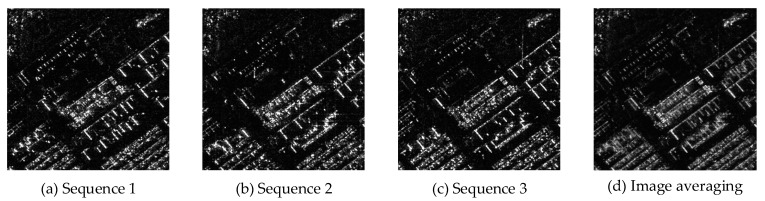
Untreated real multi-temporal SAR image (II).

**Figure 10 sensors-23-08916-f010:**
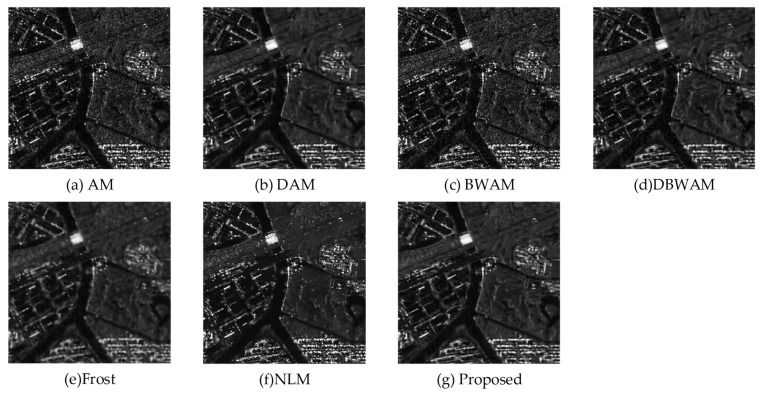
Experimental results (I).

**Figure 11 sensors-23-08916-f011:**
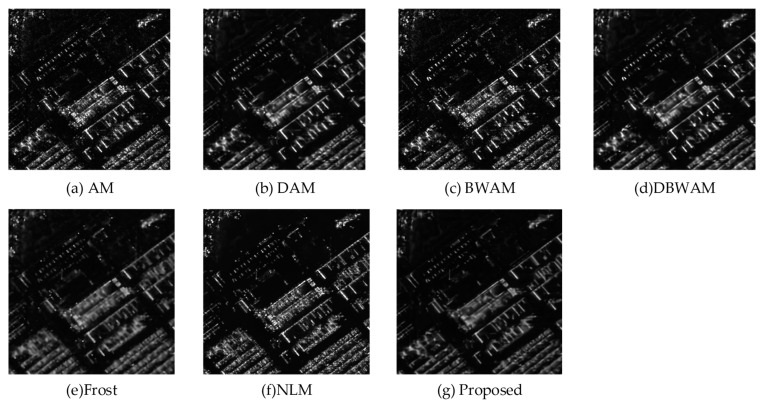
Experimental results (II).

**Figure 12 sensors-23-08916-f012:**
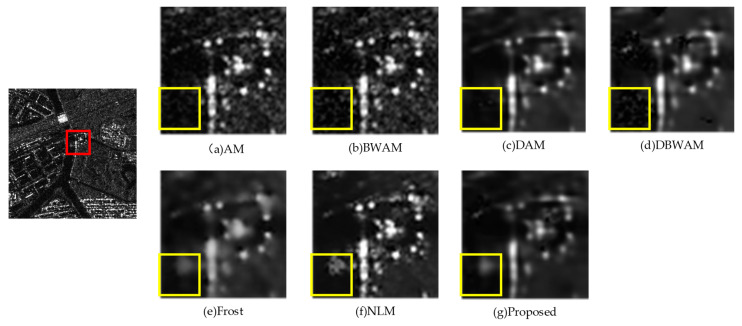
Comparison of detail amplification (I).

**Figure 13 sensors-23-08916-f013:**
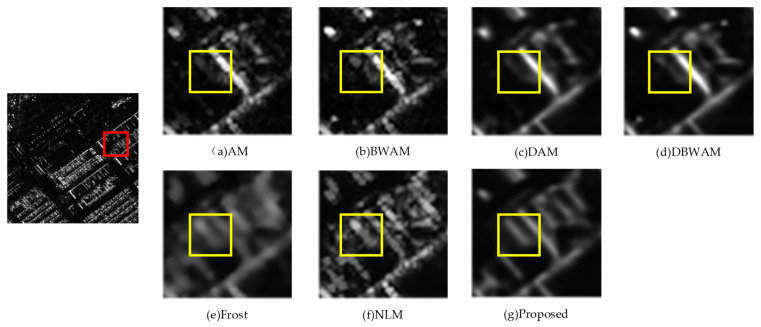
Comparison of detail amplification (II).

**Table 1 sensors-23-08916-t001:** Evaluation Metrics. (Red indicates the best, bold indicates secondary.)

**Simulated Multi-Temporal SAR Image**	**SSIM**	**Niqe**	**Corrcoef**
AM	0.3455	18.6132	0.9245
DAM	0.3455	18.6132	0.9245
BWAM	** 0.5941 **	**7.4341**	** 0.9894 **
DBWAM	** 0.5941 **	**7.4341**	** 0.9894 **
Frost	0.4885	10.4314	0.9619
NLM	0.4649	12.5985	0.9693
Proposed	**0.5534**	** 7.1388 **	** 0.9893 **

**Table 2 sensors-23-08916-t002:** Detailed parameters of the real multi-temporal SAR images.

Satellite	Orbit View	Resolution	Polarization	Incidence Angle	Region
Cosmo SkyMed	Ascending right-looking	3 m	HH	24.98°~28.35°	Shanghai

**Table 3 sensors-23-08916-t003:** The acquisition date of the SAR images.

Satellite	Sequence 1	Sequence 2	Sequence 3
Real multi-temporal SAR image (I)	2010/02/07	2010/04/12	2010/04/28
Real multi-temporal SAR image (II)	2010/02/07	2010/04/12	2010/04/28

**Table 4 sensors-23-08916-t004:** Evaluation Metrics. (Red indicates the best, bold indicates secondary.)

Real Multi-Temporal SAR Image (I)	SNR/dB	PSNR/dB	ENL	Niqe
AM	8.7484	20.4414	1.3998	6.2595
DAM	7.0705	18.7589	2.0868	6.9780
BWAM	**9.0392**	**20.7307**	1.3877	5.8263
DBWAM	7.0172	18.7055	2.1282	7.0502
Frost	2.9493	17.8931	** 2.9493 **	** 4.8454 **
NLM	1.5585	19.5814	1.5585	**5.5520**
Proposed	** 9.5634 **	** 23.7348 **	**2.1835**	6.0050
**Real multi-temporal SAR image (II)**	**SNR/dB**	**PSNR/dB**	**ENL**	**Niqe**
AM	8.2598	21.0189	0.6413	7.1949
DAM	6.8554	19.5909	0.8083	9.1173
BWAM	**8.1719**	**21.0238**	0.5916	**6.9635**
DBWAM	6.8004	19.5581	0.7784	8.6540
Frost	1.2191	17.2000	** 1.2191 **	** 5.7820 **
NLM	0.7523	18.0697	0.7523	7.4868
Proposed	** 9.2231 **	** 24.8388 **	**0.9776**	9.0078

## Data Availability

Where data is unavailable due to privacy or ethical restrictions.

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
