# Peer review of "Enhancing RABASAR for Multi-Temporal SAR Image Despeckling through Directional Filtering and Wavelet Transform"

_sensors, 2023, doi:10.3390/s23218916_

Round 1
Reviewer 1 Report
Comments and Suggestions for Authors
In this paper, the authors proposed an algorithm, named "Enhancing RABASAR for Multi-Temporal SAR Image Denoising through Directional Filtering and Wavelet Transform," to address the challenge of coherent speckle noise removal in multi-temporal SAR images. In general, the topic is very interesting. However, some comments should be well addressed before acceptance.
1. In introduction, the contribution and novelty of this paper should be highlighted. With this operation, the readers can easily follow the authors’ novelty.
2. In section 2.1.1, the directional window w is used. The reviewer wanders to know how to select this window. The authors should clearly clarify this in detail.
3. The synthetic aperture technique is also used in underwater acoustic field. The reviewer wanders to know whether the authors’ method can be used by underwater acoustic synthetic aperture technique. The authors should discuss this. These discussions would be helpful for readers in synthetic aperture sonar filed.
[1]Zhang. An omega-k algorithm for multireceiver synthetic aperture sonar. Electronics Letters. 2023, 59(13):1-3.
[2]Yang. An imaging algorithm for high-resolution imaging sonar system. Multimed Tools Appl (2023). Doi:10.1007/s11042-023-16757-0
4. In section 2.13, the Wavelet Transform is used. However, the wavelet basis is not clearly presented. That is to say, the reviewer wanders to know which wavelet basis is used by authors.
5. In section 3.2.1, the compasison between the authors’ method and traditional method should be conducted.
6. The English in this paper should be improved.
Comments on the Quality of English LanguageFine
Author Response
Dear Reviewer 1,
First and foremost, we would like to sincerely thank you for the invaluable feedback and constructive suggestions you provided for our article. Your professional insights and constructive recommendations are of paramount importance to our team.
Your feedback not only highlighted the areas where our article could be improved, but also steered us towards valuable directions for enhancement. Your contributions will directly impact the quality of our work and the credibility of our research.
We greatly value each of your suggestions and will address them with the utmost diligence in the article. We hope that, under your expert guidance, our research can achieve even more significant results.
1.In introduction, the contribution and novelty of this paper should be highlighted. With this operation, the readers can easily follow the authors’ novelty.
We deeply acknowledge this concern. We have addressed this issue by providing additional information regarding the contributions and innovative aspects of our work at the end of Section One. Kindly refer to it.
2.In section 2.1.1, the directional window w is used. The reviewer wanders to know how to select this window. The authors should clearly clarify this in detail.
We greatly appreciate your inquiry on this matter. We have supplemented equation (4) to better elucidate the selection of the target window. To illustrate with an example, if the pixel value of the central pixel is 20, and the mean pixel values of the eight segmented windows are 25, 26, 27, 28, 29, 30, 31, and 32 respectively, the computed distances are 5, 6, 7, 8, 9, 10, 11, and 12. Among these distances, the one corresponding to the closest correlation with the central pixel is 5. Therefore, we ultimately select the segmented window associated with a correlation distance of 5 as the target window.
3.The synthetic aperture technique is also used in underwater acoustic field. The reviewer wanders to know whether the authors’ method can be used by underwater acoustic synthetic aperture technique. The authors should discuss this. These discussions would be helpful for readers in synthetic aperture sonar filed.
Thank you for your suggestion, which has enriched our knowledge. Following your advice, we reviewed the relevant literature you provided. We learned that SAR operates above the atmosphere, acquiring surface information by detecting reflected microwave signals. On the other hand, SAS operates underwater, utilizing the propagation characteristics of sound waves in water. Both employ Synthetic Aperture Techniques. They enhance resolution through motion, resulting in high-quality imaging. This significantly broadens the scope of our research. In the References section of the version we are submitting, we have cited the two articles you provided. We are especially grateful.
4.In section 2.13, the Wavelet Transform is used. However, the wavelet basis is not clearly presented. That is to say, the reviewer wanders to know which wavelet basis is used by authors.
We appreciate your suggestion. Previously, our description of the wavelet transform was not sufficiently clear. We have now made the necessary revisions based on your advice. We have provided clarification on line 265. We kindly ask you to review it.
5.In section 3.2.1, the compasison between the authors’ method and traditional method should be conducted.
Regarding this section, based on your suggestions, we have conducted additional experiments in the past few days. We have included six comparative experiments for simulated SAR images. For the section involving real SAR images, we have added two comparative experiments. These include well-known traditional algorithms like Frost, as well as NLM, which is considered to perform well in the despeckling field among traditional algorithms. Additionally, in the evaluation metrics section, we have introduced three more. We hope these efforts align with your expectations. We look forward to your review.
6.The English in this paper should be improved.
We deeply acknowledge this concern. We have conducted a thorough review throughout the entire paper. Our primary focus this time was on standardizing the terminology used in the paper to ensure clarity for the readers. Additionally, we have addressed any grammar issues. We hope that this revision will make the paper much more accessible and comprehensible for your review.
Once again, we express our heartfelt gratitude for your thorough review and generous assistance.
Yours sincerely,
Jiayu

Reviewer 2 Report
Comments and Suggestions for Authors
This paper proposed an enhanced version of the RABASAR algorithm through Directional Filtering and Wavelet Transform. The enhanced RABASAR can effectively remove SAR image noise based on time-series images. However, there are still two major shortcomings in this paper: (1) The time series of three SAR images were used in the experiment, and their relevant parameters were incomplete, which made the experimental results lack credibility. (2) The performance of the enhanced RABASAR has not been further evaluated. Therefore, I recommend the authors to make a major revision to this paper. The specific comments are as follows:
Abstract
Point 1: The research background is too long. The results are short and best supported by data or digital indicators. Please rewrite the abstract.
Point 2: RABASAR is a method or framework [1], is not an algorithm.
[1] Ratio-Based Multitemporal SAR Images Denoising: RABASAR.
Introduction
Point 1: The introduction is too long and lack strict logic. For example, the author can combine and simplify the first and second paragraphs; The author should focus on the advantages and disadvantages of the algorithms (or methods) rather than describing their process. Please rewrite the introduction. I suggest the authors cite follow references:
[1] https://www.sciencedirect.com/science/article/pii/S0034425721001851.
[2] https://www.sciencedirect.com/science/article/pii/S1569843223002704
Point 2: Why did the author suddenly only mention the RABASAR framework, and why not discuss other related algorithms [1][2] ?
[1] https://ieeexplore.ieee.org/abstract/document/7948797.
[2] https://ieeexplore.ieee.org/abstract/document/6784380.
Point 3: Authors should summarize the research objective or contribution in the last paragraph of the introduction, rather than the process of solving the problem.
Research Methodology
Point 1: The process of the algorithm is explained in the method section. This process generates some intermediate variables, which makes the paper less readable. I strongly suggest that authors redraw the flowchart and list the key steps in detail.
Point 2: Why did the authors choose a 3×3 window instead of a larger one? If there are densely noise in the local area, can the 3×3 window be effectively denoised?
Point 3: Line 256-257 " We repeat this process for each pixel……". What is reference image ?? I do not find it in text, tables and figures.
Experiments
Point 1: The authors did not use a uniform name. For example, caption of figure 7(e) is "Proposed algorithm”, but caption of figure 6(b), 9(e) etc. are "Proposed". "Sequence 1" was used in Table 1, but "Sequence One" in Figure 7(a).
Point 2: In this paper, two time series containing three images are selected. However, the authors did not provide the date when SAR images were acquired. In addition, authors should compare the results of the proposed algorithm with those of other algorithms based on longer time series.
Point 3: Figures 8-13 should be redrawn and rearranged.
Point 4: Although the authors have improved the algorithm, the efficiency of the algorithm should be further evaluated. Compared to RABASAR framework, does the proposed algorithm generate the final image faster? The authors should add relevant experiments.
Comments on the Quality of English LanguageModerate editing of English language required
Author Response
Dear Reviewer 2,
We are immensely grateful for your meticulous feedback. Your expertise and constructive insights are of utmost importance to our team.
We hold each of your detailed suggestions in high regard, and we assure you that we will carefully consider and implement the corresponding revisions in the paper. Under your professional guidance, we are confident that our research will achieve even more significant results.
Abstract
Point 1: The research background is too long. The results are short and best supported by data or digital indicators. Please rewrite the abstract.
Point 2: RABASAR is a method or framework, is not an algorithm.
We greatly appreciate your suggestions. In response, we have reworked the abstract section by reducing the background information. Additionally, we have provided a more detailed description in the results section. Regarding the nature of RABASAR, it is a method or a framework, we now have a clearer understanding. Correspondingly, we have made the necessary adjustments throughout the entire manuscript.
Introduction
Point 1: The introduction is too long and lack strict logic. For example, the author can combine and simplify the first and second paragraphs; The author should focus on the advantages and disadvantages of the algorithms (or methods) rather than describing their process. Please rewrite the introduction. I suggest the authors cite follow references:
[1] https://www.sciencedirect.com/science/article/pii/S0034425721001851.
[2] https://www.sciencedirect.com/science/article/pii/S1569843223002704
Point 2: Why did the author suddenly only mention the RABASAR framework, and why not discuss other related algorithms [1][2] ?
[1] https://ieeexplore.ieee.org/abstract/document/7948797.
[2] https://ieeexplore.ieee.org/abstract/document/6784380.
Point 3: Authors should summarize the research objective or contribution in the last paragraph of the introduction, rather than the process of solving the problem.
We have implemented all of your suggestions in the introduction section. We have significantly reduced the description of the algorithm process, and merged the first and second paragraphs. We have thoroughly reviewed the entire introduction and have made extensive reductions to streamline the paper, resulting in a more concise presentation. In lines 35 to 39, we have incorporated the references you provided, which has enriched the discourse of our paper. Following your suggested literature, we discussed various other multi-temporal despeckling methods. In the final paragraph of the introduction, we added a statement about the contributions of our research. Your suggestions have been instrumental in enhancing the quality of our manuscript.
Research Methodology
Point 1: The process of the algorithm is explained in the method section. This process generates some intermediate variables, which makes the paper less readable. I strongly suggest that authors redraw the flowchart and list the key steps in detail.
Point 2: Why did the authors choose a 3×3 window instead of a larger one? If there are densely noise in the local area, can the 3×3 window be effectively denoised?
Point 3: Line 256-257 " We repeat this process for each pixel……". What is reference image ?? I do not find it in text, tables and figures.
Regarding your point 1, we have redrawn the flowchart. In the flowchart, we have included detailed descriptions of DSMT-NLM. Additionally, for better readability, we have incorporated necessary equations, such as the ratio part, and added some essential symbols. We hope you will review it.
Regarding your point 2, we want to express our special gratitude. When our team was designing this framework, we had a thorough discussion on this aspect. We tried different window sizes including 3x3, 5x5, and 7x7. However, in our subsequent experiments, we found that the despeckling effects of 5x5 and 7x7 windows were not satisfactory. Therefore, we ultimately opted for the 3x3 window.
Regarding your point 3, we have made the necessary adjustments. We realized that the use of the term C in the entire narrative seemed somewhat abrupt and difficult to grasp. Therefore, in the revised flowchart and subsequent wavelet transform, we deliberately incorporated this symbol repeatedly to ensure that readers can understand it clearly.
Experiments
Point 1: The authors did not use a uniform name. For example, caption of figure 7(e) is "Proposed algorithm”, but caption of figure 6(b), 9(e) etc. are "Proposed". "Sequence 1" was used in Table 1, but "Sequence One" in Figure 7(a).
Point 2: In this paper, two time series containing three images are selected. However, the authors did not provide the date when SAR images were acquired. In addition, authors should compare the results of the proposed algorithm with those of other algorithms based on longer time series.
Point 3: Figures 8-13 should be redrawn and rearranged.
Point 4: Although the authors have improved the algorithm, the efficiency of the algorithm should be further evaluated. Compared to RABASAR framework, does the proposed algorithm generate the final image faster? The authors should add relevant experiments.
For point 1, we appreciate your meticulous work. We have made the necessary revisions by standardizing the naming convention. Additionally, we have addressed the same issue throughout the entire manuscript and made the necessary corrections.
Regarding your point 2, we have added the acquisition dates of the SAR images, as shown in Table 3. However, unfortunately, our team currently has limited SAR data, with a maximum of only three time-series SAR images. We also reached out to other relevant researchers, but they declined to provide us with additional time-series SAR data. We sincerely apologize for this limitation and hope for your understanding.
Regarding your point 3, we have rearranged Figures 8-13. Additionally, we have made adjustments to other images, including explanatory diagrams. Once again, we sincerely appreciate your suggestions.
Regarding your point 4, we greatly appreciate your feedback. Our team has indeed discussed and attempted to optimize the algorithm's efficiency during the experimental process. However, due to our use of DSMT-NLM, which involves traversing both locally and globally, this operation does improve our despeckling effect but does increase our algorithm's runtime. Indeed, this may be one of the challenges in balancing local and global filtering. On the other hand, the original RABASAR obtains the “superimage” through weighted averaging. While this method is more efficient, it results in the loss of a significant amount of fine detail information from the multi-temporal SAR images, which contradicts the initial purpose of using multi-temporal SAR to increase information content. Our team's next plan is to enhance the efficiency of this algorithm while ensuring the despeckling effect.
Once again, we deeply appreciate your time and patience. We look forward to your valuable feedback on our revisions.
Warmest regards,
Jiayu

Round 2
Reviewer 1 Report
Comments and Suggestions for Authors
The authors well revise the paper.
Comments on the Quality of English LanguageFine
Reviewer 2 Report
Comments and Suggestions for Authors
All my concerns have been revised, I recommend to accept this paper on currrent version.
Comments on the Quality of English LanguageModerate editing of English language required